# Organokine-Mediated Crosstalk: A Systems Biology Perspective on the Pathogenesis of MASLD—A Narrative Review

**DOI:** 10.3390/ijms262311547

**Published:** 2025-11-28

**Authors:** Sandra Maria Barbalho, Lucas Fornari Laurindo, Vitor Engracia Valenti, Nahum Méndez-Sánchez, Mariana M. Ramírez-Mejía, Ricardo de Alvares Goulart

**Affiliations:** 1Department of Biochemistry and Pharmacology, School of Medicine, Universidade de Marília (UNIMAR), Marília 17525-902, SP, Brazil; 2Postgraduate Program in Structural and Functional Interactions in Rehabilitation, School of Medicine, Universidade de Marília (UNIMAR), Marília 17525-902, SP, Brazil; ricardogoulartmed@hotmail.com; 3Department of Biochemistry and Nutrition, School of Food and Technology of Marília (FATEC), Marília 17500-000, SP, Brazil; 4Department of Research, Research Coordination Center, UNIMAR Charitable Hospital, Universidade de Marília (UNIMAR), Marília 17525-902, SP, Brazil; 5Laboratory for Systematic Investigations of Diseases, Department of Biochemistry and Pharmacology, School of Medicine, Universidade de Marília (UNIMAR), Marília 17525-902, SP, Brazil; 6Division of Cellular Growth, Hemodynamic, and Homeostasis Disorders, Graduate Program in Medical Sciences, Faculdade de Medicina, Universidade de São Paulo (USP), São Paulo 01246-903, SP, Brazil; 7Faculty of Philosophy and Sciences, Universidade Estadual Paulista (UNESP), Campus Marília, Marília 17525-900, SP, Brazil; vitor.valenti@unesp.br; 8Liver Research Unit, Medica Sur Clinic & Foundation, Mexico City 14050, Mexico; nmendez@medicasur.org.mx (N.M.-S.); mich.rm27@gmail.com (M.M.R.-M.); 9Faculty of Medicine, National Autonomous University of Mexico, Mexico City 04510, Mexico; 10Plan of Combined Studies in Medicine (PECEM-MD/PhD), Faculty of Medicine, National Autonomous University of Mexico, Mexico City 04510, Mexico

**Keywords:** MASLD, liver disease, organokines, inflammation, insulin resistance

## Abstract

Metabolic dysfunction-associated steatotic liver disease (MASLD) is a prevalent chronic condition with a complex pathophysiology involving multiple organs. Organokines, including hepatokines, myokines, cardiokines, renokines, osteokines, and adipokines, play central roles in lipid metabolism, glucose homeostasis, inflammation, and fibrosis. Dysregulation of these signaling molecules contributes to the progression of MASLD and its systemic complications. This review examines the role of organokine-mediated crosstalk between the liver and peripheral organs (e.g., muscle, heart, kidneys, bone, and adipose tissue) in the pathogenesis of MASLD. Key molecules, such as myostatin, FGF-21, IL-6, and adiponectin, influence insulin sensitivity, lipid metabolism, and inflammation. Some organokines have protective effects (e.g., FGF-21, irisin, and klotho), while others, such as myostatin and fetuin-A, exacerbate insulin resistance and fibrosis. These findings suggest that targeting organokines could provide potential biomarkers and therapeutic strategies for MASLD. Future research should focus on elucidating the molecular mechanisms and assessing the role of organokines in the prevention and treatment of MASLD.

## 1. Introduction

People’s life expectations have increased thanks to the many benefits of modern society. However, an inadequate lifestyle can have a profound impact on people’s health and quality of life. Increased consumption of fats and sugar and lack of physical exercise can be related to overweight/obesity, insulin resistance/diabetes, dyslipidemia, metabolic syndrome, and cardiovascular diseases (CVD). On the other hand, these conditions are associated with increased inflammatory processes, oxidative stress, and several other factors that can exacerbate existing diseases or lead to the development of others, such as liver diseases and sarcopenia [1,2,3,4,5,6]. In 1986, the medical term non-alcoholic fatty liver disease, also known as NAFLD, was coined for a condition that would become the most prevalent liver disorder worldwide. However, over the last 3–4 years, specialists from around the world have come together to remove NAFLD from the medical dictionary and propose a new nomenclature for the disease, given the global rise in the development of cardiometabolic risk factors [7].

Metabolic dysfunction-associated steatotic liver disease (MASLD) is a prevalent liver condition worldwide, affecting over 38.0% of adults globally; however, it is projected to reach approximately 56% in the next decade [8,9,10]. It is the second-leading cause for liver transplantation (and already the leading cause in women) worldwide, and significantly affects the population’s health [11,12,13]. The diagnosis includes liver steatosis, mainly associated with obesity, type 2 diabetes, dyslipidemia, and high blood pressure [14,15,16,17,18,19]. The definition of MASLD was proposed to include steatosis in the role scenario of the patient’s condition [20]. MASLD encompasses hepatic steatosis and more advanced features, including inflammation, hepatocyte ballooning, and lipoapoptotic damage. This condition can evolve from simple steatosis to cirrhosis and hepatocellular carcinoma [21].

Moreover, MASLD has emerged as the most important cause of chronic liver disease on the globe and, as pointed out above, has been driven mainly by the global epidemic of type 2 diabetes mellitus and obesity [6]. Insulin resistance and abdominal fat are potent factors in the development of MASLD. Since lipid metabolism is affected, there is an increase in inflammatory processes, oxidative stress, and mitochondrial dysfunction, which interfere with lipid metabolism and promote hepatic fat deposition [16,22,23,24].

Unfortunately, MASLD is not limited to Western countries. Studies have shown that it is also increasing in regions regularly associated with viral hepatitis and leading to a dual burden of MASLD and hepatitis B virus infection, contributing to the progression of increased risk for liver diseases and liver cancer [6,15,25,26].

A way to reduce the burden of MASLD requires a multidisciplinary approach involving public health professionals, policymakers, medical specialists, and other relevant experts [27]. This underscores the importance of raising awareness about risk factors, promoting early diagnosis, and advancing innovative public health policies. Properly trained physicians play a critical role in the early identification of liver diseases and in initiating effective therapeutic interventions. When supported by well-structured public policies, such efforts can mitigate the progression of disease and reduce the incidence of severe complications, including cirrhosis and hepatocellular carcinoma [28,29,30,31,32]. These actions could reduce public and private costs with the specific and expensive treatment of these conditions. Other factors that may be related to the burden on health systems are premature mortality and the inability to work [27].

Understanding the mechanisms behind MASLD requires a multifaceted approach, with significant contributions from both preventive and therapeutic research. A key area of focus is the crosstalk between the liver and other organs, which provides insight into how interorgan interactions can either exacerbate or alleviate the condition. The liver engages with skeletal and cardiac muscles, kidneys, adipose tissue, and bone through various chemical mediators known as organokines. These signaling molecules play a crucial role in modulating the progression of MASLD. In this review, we first summarize foundational literature on organokine-mediated communication, followed by recent advances in the field that shed light on their role in MASLD pathogenesis. We then identify critical gaps in current knowledge and highlight the potential for targeting these pathways therapeutically. Specifically, we focus on how hepatokines, myokines, adipokines, cardiokines, renokines, and osteokines influence key processes, including insulin sensitivity, lipid metabolism, inflammation, and fibrosis. We also discuss emerging biomarkers and intervention strategies based on this interorgan network, which could offer new diagnostic and therapeutic opportunities. The methodology of this review is described in Table A1 in Appendix A.

## 2. Exploring the Liver-Muscle Crosstalk (Hepatokines and Myokines) in MASLD

Many authors have demonstrated the association between hepatic and muscular metabolism [33,34,35,36]. Depending on the secretory pattern, this liver-muscle axis has been shown to have numerous metabolic repercussions and exacerbate liver diseases [16,37].

Skeletal muscle plays a fundamental role not only in locomotion and postural control but also in systemic metabolic regulation. It is a significant site for glucose uptake and utilization, thereby contributing to the maintenance of normoglycemia, and serves as a critical reservoir of amino acids during fasting. Consequently, metabolic and structural alterations in skeletal muscle can disrupt insulin signaling, promote chronic low-grade inflammation, and increase oxidative stress, all of which are closely associated with the pathogenesis of MASLD [38,39]. Furthermore, muscle has been considered a proper endocrine organ, releasing myokines that can have systemic effects. These myokines can have diverse effects and are related to antioxidant and anti-inflammatory properties when muscle homeostasis is maintained; however, they can also be associated with harmful impacts when muscle mass undergoes undesirable changes [40,41].

Myokines and hepatokines engage in crosstalk through the endocrine, paracrine, and autocrine pathways. Notwithstanding, they have specific associations regarding obesity, glucose metabolism, insulin resistance, diabetes mellitus, and metabolic syndrome [42,43,44]. Therefore, these biomarkers have been proposed to aid in the diagnosis and prognosis of liver, muscle, and systemic conditions, thereby innovating therapeutic approaches [45,46,47]. Table 1 presents the primary hepatokines and myokines associated with MASLD.

An essential consequence of metabolic impairment in muscle can result in sarcopenia, a progressive disorder that has increased worldwide. Following the European Working Group on Sarcopenia in Older People (EWGSOP), the prevalence of this condition is 8–36% in subjects under 60 years old and 10–27% in those aged 60 years and above [95,96,97]. It affects skeletal muscles, resulting in loss of muscle mass, strength, and function [98,99,100,101,102]. This condition is commonly associated with the aging process, hospitalization, oxidative stress, inflammation, diabetes/insulin resistance, as well as mitochondrial dysfunction, and liver dysfunction, such as MASLD [103,104], leading to physical dysfunction, impaired or severely diminished quality of life, increased morbidity, and mortality [105,106].

The physiopathological mechanisms related to MASLD and sarcopenia remain incompletely understood. As already mentioned, insulin resistance is a significant risk factor for MASLD and plays a crucial role in the muscle-liver crosstalk. People with reduced muscle strength have elevated inflammatory markers such as Interleukin (IL)-1β, IL-6, Tumor Necrosis Factor-α (TNF-α), and high-sensitivity C-reactive Protein (hs-CRP), suggesting the close relationship between muscle mass impairment and liver diseases [107,108,109,110,111]. A notable example is the release of TNF-α, which is related to the physiopathology of MASLD due to the pro-inflammatory macrophages that develop during adipocyte hypertrophy, leading to lipolysis (and an increase in free fatty acids) and favoring insulin resistance [22,112]. Insulin resistance, driven by elevated levels of pro-inflammatory cytokines, contributes to skeletal muscle catabolism. Additional shared risk factors linking muscular and hepatic dysfunction include dyslipidemia, obesity, type 2 diabetes mellitus, and metabolic syndrome. All of which play central roles in the pathophysiology of both sarcopenia and metabolic-associated liver diseases [113,114,115,116].

Furthermore, it is relevant to consider the secretory pattern of the skeletal muscle. Myokines have essential roles in the balance of glucose and lipid metabolism. Impaired muscle mass can also impair the release of myokines [35,71,117,118,119,120]. On the other hand, patients with MASLD can alter myokine levels, exposing muscles to increased ammonia concentrations resulting from hepatocyte dysfunction and an imbalanced urea cycle [110,121]. In addition, fibroblast growth factor-21 (FGF-21) levels are higher in decompensated cirrhosis patients presenting sarcopenia. FGF-21 functions as a liver-muscle crosstalk, imbalances muscle mass regeneration by inhibiting the PI3K/Akt (Phosphoinositide 3-kinase/Protein kinase b) pathways [16,110,122]. The liver and muscle share many metabolic and physiological pathways; thus, MASLD and muscle impairment share the consequences of hepatokines, myokines, insulin resistance, oxidative stress, and chronic inflammation. For these reasons, it is possible to conclude that one disease contributes to the development of the other. Apart from the physiopathological relationship, it is essential to understand the clinical connection between MASLD and sarcopenia to identify high-risk patients and provide effective clinical management. The early recognition of the relationship between muscle and liver diseases can improve the patient’s prognosis [14,107,123,124].

In a meta-analysis, the authors demonstrated that sarcopenia increases the risk of MASLD and is associated with an elevated risk of liver fibrosis [125]. For those mentioned reasons, sarcopenia is not only considered an aging-related syndrome, but it is acknowledged to be secondarily related to chronic diseases such as MASLD [105,120,126,127,128,129,130,131,132].

The role of hepatokines (that directly modulate hepatic homeostasis) in the context of MASLD includes Angiopoietin-Like 3 (ANGPTL3), that impairs lipid clearance, being associated with dyslipidemia and steatosis; activin protects against fatty acid influx into the liver (beneficial regulator); fetuin-A promotes insulin resistance and hepatic lipid accumulation; FGF-21 (hepatic) increases fatty acid oxidation, with a protective role in MASLD. Additionally, selenoprotein P promotes insulin resistance and liver inflammation. Ultimately, lower levels of sex hormone-binding globulin (SHBG) increase the risk of MASLD [50,51,53,54,55,133,134,135].

In summary, hepatokines such as FGF-21, fetuin-A, ANGPTL3, and others play key roles in maintaining hepatic and systemic metabolic homeostasis. Their dysregulation contributes to lipotoxicity, insulin resistance, and inflammation, central events in the pathogenesis of MASLD. A deeper understanding of hepatokine signaling provides new insights into the systemic nature of the disease, highlighting these molecules as potential diagnostic biomarkers and therapeutic targets.

On the other hand, myokines serve as essential mediators of muscle-liver crosstalk, especially in the context of physical activity and metabolic health. Alterations in molecules such as irisin and IL-6 can either exacerbate or attenuate hepatic steatosis, depending on the physiological context. These findings underscore the protective effects of skeletal muscle homeostasis and supportive exercise as a non-pharmacological strategy in the prevention and management of MASLD.

## 3. Exploring the Liver-Heart Crosstalk (Hepatokines and Cardiokines) in MASLD

As previously noted, clinical and pathophysiological evidence support a causal association between MASLD and CVD [136,137]. MASLD plays a pivotal role in the development and progression of CVD, often preceding hepatic complications such as cirrhosis or hepatocellular carcinoma. The heart actively contributes to this pathophysiological interplay through the secretion of bioactive molecules collectively referred to as cardiokines [11,138].

In MASLD, the steatotic scenario and an environment of lipotoxicity impair liver regulatory functions [139], leading to the accumulation of triglycerides within hepatocytes and the release of atherogenic lipoproteins into the circulation. The increase in liver lipids leads to the release of pro-inflammatory cytokines, such as IL-6 and TNF-α, and the production of reactive oxygen species (ROS), contributing to systemic inflammation, vascular damage, and accelerated atherosclerosis [140,141]. Endothelial dysfunction, a hallmark of early CVD, is commonly observed in MASLD and is exacerbated by both hyperinsulinemia and oxidative stress [142]. This dysfunction impairs nitric oxide bioavailability, leading to increased vascular tone, platelet aggregation, and ultimately, the formation and instability of plaques. Additionally, MASLD is associated with coronary artery calcification, a validated surrogate of coronary atherosclerosis and predictor of major cardiovascular events [143,144,145].

In a prospective five-year study, the authors have demonstrated that the increased risk for CVD in type-2 diabetic participants was partly associated with metabolic syndrome, but significantly linked to NAFLD [146]. A meta-analysis with 34,043 participants demonstrated that patients with NAFLD have an elevated risk of fatal and non-fatal cardiovascular events [147,148]. Moreover, liver inflammation, histologic severity, and the MASLD activity score were correlated with the severity of inflammation in adipose tissue, characterized by increased pro-inflammatory macrophages, TNF-α, and IL-6 in the visceral adipose tissue [11,138].

Some authors have demonstrated that myocardial ischemia, especially acute myocardial infarction, exacerbates liver injury in metabolic-associated fatty liver disease (MAFLD) through two central and interrelated mechanisms. The first is related to the elevation of circulating inflammatory monocytes and their accelerated recruitment to the liver via the MCP-1 (monocyte chemoattractant protein-1)/CCR2 (MCP-1 receptor) axis, and the second involves the synthesis and release of periostin, an extracellular matrix protein re-expressed by cardiac fibroblasts during cardiac injury [149,150,151].

In summary, cardiokines can profoundly influence liver metabolism and vice versa. Atrial Natriuretic Peptide (ANP) regulates liver glycogen metabolism and glucose homeostasis. B-Type Natriuretic Peptide (BNP) is a biomarker of volume overload and is increased in cirrhosis. Growth Differentiation Factor 15 (GDF-15) is involved in the heart-liver crosstalk in cardiac pathogenesis. IL-33 can reduce the inflammatory process as well as hepatic fibrosis in early stages. Cardiac myostatin is associated with muscle atrophy and hepatic fibrosis. On the other hand, natriuretic peptides improve lipid metabolism and may play a protective role against MASLD [54,66,67,71,72,73]. Table 1 summarizes the effects of some cardiokines related to MASLD, and Figure 1 shows the relationship between the liver and the heart.

In conclusion, cardiokines represent a crucial link in the bidirectional communication between the liver and the cardiovascular system. In MASLD, these molecules may contribute to hepatic fibrogenesis and insulin resistance, while the progression of liver disease can, in turn, exacerbate cardiac remodeling and endothelial dysfunction. This reciprocal interaction highlights the significance of cardiovascular risk assessment in patients with MASLD, suggesting cardiokines as potential biomarkers and therapeutic targets for early intervention and risk stratification.

## 4. Exploring the Liver-Kidney Crosstalk (Hepatokines and Renokines) in MASLD

The kidneys and liver maintain complementary functions, thus playing a central role in maintaining homeostasis and detoxification. Specifically in the kidneys, the regulation of blood volume/pressure, electrolyte and acid-base balance, secretion of erythropoietin and calcitriol, prostaglandin synthesis, and control of the renin–angiotensin–aldosterone system (RAAS) [152]. Failure of these mechanisms increases morbidity and mortality, triggering multisystem dysfunctions with a significant impact on the liver [153,154,155,156].

The regulation of renal function involves a complex network of signaling molecules, known as renokines, which influence systemic homeostasis and the progression of kidney disease [157]. These may include Bone Morphogenetic Proteins (BMPs), particularly BMP-7, which antagonizes the pro-fibrotic TGF-β pathway, inhibiting epithelial–mesenchymal transition and attenuating renal fibrosis and inflammation [158,159].

Erythropoietin (EPO) is a glycoprotein hormone primarily synthesized in the kidneys, which is essential for erythropoiesis. In chronic kidney disease (CKD), its reduced production leads to normocytic, normochromic anemia, exacerbating fatigue and cardiovascular dysfunction [160].

Osteocytes produce Fibroblast Growth Factor 23 (FGF-23) in response to hyperphosphatemia. In early CKD, it promotes renal phosphate excretion, but its sustained elevation contributes to ventricular hypertrophy, cardiovascular calcifications, and kidney disease progression [161,162].

GDF-15, a member of the transforming growth factor-beta (TGF-β) superfamily, is stimulated by cellular stress, including stress in mitochondria, endoplasmic reticulum, and liver. It is also released with exercises, smoking, and age [157]. Klotho is a transmembrane protein with anti-aging action, expressed in renal tubules. It acts as a co-receptor for FGF-23, regulating phosphate and vitamin D metabolism. Its depletion in CKD is associated with vascular calcification, renal fibrosis, and increased mortality [163,164].

Lipocalin-2 is a biomarker of acute kidney injury with two functions: at physiological levels, it promotes tissue repair; when persistently elevated in CKD, it exacerbates tubular inflammation and cellular apoptosis [165].

The RAAS is involved in the chronic activation of renin and angiotensin II, which induces vasoconstriction, glomerular inflammation, and interstitial fibrosis, thereby worsening hypertension and kidney injury [166,167].

TNF-Related Inducer of Apoptosis is a multifunctional cytokine that, at high concentrations, activates inflammatory pathways such as Nuclear Factor Kappa B (NF-κB) and promotes glomerular fibrosis, correlating with the progression of CKD. TNF-α is also produced by T cells and resident kidney cells (such as mesangial cells, renal epithelial cells, and vascular endothelial cells) [168]. It can also induce a natriuretic effect and stimulate hypertensive response and renal injury, which is also toxic to glomerular epithelial cells. IL-1 is produced by hematopoietic cells and resident kidney cell lineages. IL-1α and IL-1β bind to the same receptor for IL-1. The NLRP3 inflammasome mediates the conversion of pro-IL-1β to its active form. The activation of its receptor leads to the recruitment of Myd88 and many IL-1 receptor-associated kinases, inducing the translocation of NF-κB’s p65 component to the nucleus, which drives the transcription of genes related to inflammatory proteins, such as TNF [156,169].

Kidney-liver crosstalk is essential in normal and some pathological conditions. Renal-induced liver harm and liver-induced kidney diseases can occur due to the interaction between these organs [170]. This phenomenon is associated with cytokine synthesis and the activation of pro-inflammatory pathways, as well as oxidative stress, ischemia and reperfusion, metabolic acidosis, and modifications in enzyme activity. These metabolic pathways provide the foundation for the interaction between the kidneys and liver [156].

Regarding cytokine production, acute renal failure triggers a low-grade systemic inflammatory response characterized by elevated levels of cytokines, such as IL-6, IL-17A, and TNF-α. These mediators promote endothelial dysfunction and increased hepatic vascular permeability, facilitating immune cell infiltration and liver injury. This scenario impairs cytochrome P450 (CYP) enzyme activity, mainly CYP3A4, CYP3A1/2, and CYP2E1, thereby compromising hepatic drug metabolism. Notwithstanding, indoxyl sulfate accumulation and increased IL-10 levels further suppress CYP function. IL-6 also induces the synthesis of acute-phase proteins while downregulating hepatic proteins such as albumin [156,171].

In the case of oxidative stress, studies have shown that both ischemic and non-ischemic acute renal failure trigger oxidative stress, contributing to liver injury. The early oxidative activation leads to leukocyte infiltration, hepatocyte apoptosis, and depletion of hepatic antioxidants, such as superoxide dismutase and catalase, while increasing lipid peroxidation markers, including malondialdehyde [172,173].

The liver and kidneys are crucial modulators of acid-base balance. In chronic renal failure, the balancing mechanism can be disrupted, leading to metabolic acidosis [174]. Acute acidosis profoundly interferes with the body. One consequence can be a disturbance in glutamine metabolism, resulting in an acute increase in blood glutamine concentrations and a decline in total hepato-splanchnic glutamine concentrations. Some authors have demonstrated that chronic acidosis reduces glutamine extraction and albumin synthesis in the liver, while increasing renal glutamine and ammonia production [175]. Furthermore, acidosis is linked to an increased rate of systemic protein turnover (both protein synthesis and proteolysis) [176,177,178].

On the other hand, in liver failure, for example, in cirrhosis state, its multisystem complications include acute kidney injury in part of the hospitalized patients [179,180]. Hepatorenal syndrome, a severe form of acute kidney disease in advanced cirrhotic patients, is characterized by renal vasoconstriction and decreased glomerular filtration rate. Its mechanism involves portal hypertension, splanchnic vasodilation, neurohormonal activation (RAAS/sympathetic), and renal hypoperfusion [181]. Biglycan (a protein released by the cirrhotic liver) promotes renal inflammation via TLR2/TLR4 and autophagy [49,182]. Cirrhotic cardiomyopathy and adrenal insufficiency exacerbate renal hypoperfusion, and relative adrenal deficiency further exacerbates renal hypoperfusion [57,183,184].

In summary, the role of renin influences hormonal regulation and renal-hepatic stress. Some examples include erythropoietin, which, among other functions, modulates insulin sensitivity and may reduce liver steatosis. Klotho exhibits anti-inflammatory and antioxidant properties, which contribute to a protective effect in MASLD. Neutrophil gelatinase-associated lipocalin (NGAL or lipocalin 2) is a biomarker of renal stress and correlates with the severity of liver injury. Finally, renin is associated with the activation of the RAAS axis and worsening of steatohepatitis [74,76,77,78,79].

In conclusion, renokines play a significant role in the systemic manifestations of MASLD. Some of them are related to modulating oxidative stress, vascular tone, and hepatic function. In metabolic disease states, kidney dysfunction can lead to the accumulation of uremic toxins and reduced clearance of inflammatory mediators, contributing to hepatic injury and fibrosis. Furthermore, MASLD itself may exacerbate renal damage through systemic inflammation and altered lipid metabolism, establishing a vicious cycle. Recognizing the liver-kidney axis in MASLD pathophysiology supports the development of integrated therapeutic strategies aimed at preserving renal function and mitigating hepatic progression. Figure 2 summarizes the relationship between the liver and the kidney. Table 1 presents the primary renokines associated with MASLD.

## 5. Exploring the Liver-Adipose Tissue Crosstalk (Hepatokines and Adipokines) in MASLD

Adipokines are biomarkers produced by adipose tissue that regulate glucose and lipid metabolism, influencing liver diseases, appetite, insulin sensitivity, and fibrogenesis [185]. Under normal physiological conditions, adipokines exert regulatory effects on multiple organs, including the brain, liver, skeletal muscle, bone, and the endocrine pancreas [186,187].

In overnutrition, adipose tissue undergoes structural and functional remodeling, resulting in alterations in the secretion of adipokines and inflammatory mediators. This uncontrolled pattern, especially involving adipokines with metabolic and immunoregulatory functions, contributes to adipose tissue dysfunction [188]. This imbalance alters both the secretion profiles and biological actions of adipokines, promoting an increase in ROS, leading to endoplasmic reticulum stress and mitochondrial dysfunction [189,190]. The consequences lead to a state of chronic low-grade inflammation, exacerbating metabolic dysregulation and insulin resistance [191]. In addition to the increase in ROS, adipose tissue also leads to an imbalance of pro-inflammatory adipocytokines, such as IL-6, TNF-α, and resistin [192,193]. In this scenario, there is an amplification of reactive species production, inflammation, and metabolic injury. Another consequence is that the lipid storage capacity of adipose tissue is compromised, favoring the diversion of excess fatty acids to other tissues such as the liver, where their ectopic deposition promotes lipotoxicity and metabolic disorders [188,190,191,194].

Emerging data indicates a sophisticated, bidirectional communication axis between adipose tissue and the liver, coordinating systemic energy homeostasis. This interplay, mediated by both hepatokines and adipokines, plays a central role in the onset and progression of MASLD [193,195].

Among adipokines, adiponectin exerts protective effects by activating the AMPK and PPAR-α pathways, thereby reducing hepatic steatosis, inflammation, and fibrosis; notably, its levels decline as MASLD progresses. In contrast, leptin plays a role in early metabolic regulation; however, chronic elevations and leptin resistance contribute to the worsening of hepatic inflammation and disease progression [196,197]. In obese individuals, adipose tissue hypertrophy leads to a state of chronic low-grade inflammation, increases the attraction of chemotactic molecules, and decreases adiponectin levels. Furthermore, this inflammatory state significantly contributes to insulin and leptin resistance [42,198].

The FGF-21-adiponectin axis serves as a key mediator of crosstalk between the liver and adipose tissue, exerting beneficial effects on maintaining energy consumption homeostasis [199,200].

RBP4 (Retinol-binding protein 4) is an adipokine that is secreted primarily by hepatic and adipose cells and has a role in transporting retinol from the liver to peripheral tissues. Some studies have shown that the levels of this adipokine are associated with an increased risk of metabolic diseases, including obesity, type 2 diabetes, and MASLD [201,202].

Other adipokines, such as resistin, visfatin, and chemerin, also modulate hepatic lipid handling and inflammatory responses, although their roles are less clearly defined. This bidirectional liver-adipose axis forms a feedback loop where hepatokine/adipokine imbalances exacerbate insulin resistance and hepatic lipid deposition [203,204,205,206,207].

Ultimately, the effects of numerous adipokines have been known for a long time. In summary, adiponectin plays a significant anti-inflammatory role, in addition to improving insulin sensitivity. It also has protective effects against steatosis and consequent fibrosis [208,209]. Leptin acts on the hypothalamic satiety center and is associated with decreased inflammation and steatosis. However, obese individuals may develop hyperleptinemia and hepatic fibrosis, thus exhibiting unbeneficial effects. Leptin is also increased in MASLD. Resistin, as its name suggests, induces insulin resistance and is associated with the development of steatosis. Another molecule associated with insulin resistance is visfatin. It stimulates NF-κB, consequently increasing pro-inflammatory cytokines such as TNF-α, IL-6, and IL-1β. It is associated with the worsening of steatosis. RBP4 levels are related to increased risk of metabolic diseases, including obesity, type 2 diabetes, and MASLD. The FGF-21-adiponectin axis is a key mediator of crosstalk between the liver and adipose tissue, exerting beneficial effects [85,200,210,211,212,213,214,215]. Table 1 illustrates the impact of some adipokines, and Figure 3 summarizes the relationship between liver and white adipose tissue.

Ultimately, adipose tissue, particularly in its dysfunctional and inflamed state, acts as a critical endocrine organ through the secretion of adipokines that modulate hepatic lipid metabolism and inflammation. The imbalance between pro-inflammatory and anti-inflammatory adipokines is closely associated with MASLD progression. These insights underscore the significance of adipose-liver communication and reinforce the therapeutic potential of targeting adipokine pathways in the treatment of metabolic liver disease.

## 6. Exploring the Liver-Bone Tissue Crosstalk (Hepatokines and Osteokines) in MASLD

Although a link between bone and liver seems unlikely, a bidirectional liver-bone axis exists, in which bone-derived osteokines regulate hepatic metabolism and contribute to the pathogenesis of MASLD [216]. For these reasons, bone is also recognized as an endocrine organ, secreting osteokines such as FGF-23, NGAL, osteopontin, and osteocalcin that interact with the liver. These osteokines can predict the incidence and progression of metabolic disorders and liver diseases [217,218].

Protective osteokines include osteocalcin, which reduces liver steatosis. Conversely, other osteokines, such as osteopontin, which is elevated in MASLD, promote inflammation and fibrosis. FGF-23, which is associated with increased insulin resistance, and sclerostin, which reduces insulin sensitivity, exacerbate the progression of MASLD and its consequences [219,220].

In summary, osteokines such as FGF-23 correlate with vascular calcification and endothelial dysfunction, in addition to being associated with an increased risk of steatosis and portal hypertension. Osteocalcin promotes hepatoprotective effects. In animal models, its administration activates the Nuclear factor erythroid 2-related factor 2 (Nrf2) pathway [209], increases the expression of antioxidant enzymes such as superoxide dismutase, catalase, and glutathione peroxidase, and inhibits cJun N-terminal kinase (JNK) activation, resulting in reduced steatosis and liver injury [221]. Furthermore, it regulates fatty acid uptake via CD36 through the AMPK/FOXO1/BCL6 pathway [222]. It is related to increased insulin release and reduced fibrosis in MASLD [223,224,225,226]. Osteopontin promotes hepatic inflammation and fibrosis, and its levels are increased in MASLD/hepatic steatosis. Its levels are elevated in patients with Nonalcoholic steatohepatitis (NASH), and promote macrophage recruitment and activation of fibrogenic pathways in the liver [227,228]. Finally, sclerostin, produced by osteocytes, interferes with Wnt/β-catenin signaling, being associated with metabolic dysfunction, insulin resistance, and MASLD [93]. NGAL upregulation may work as a protective osteokine, reducing obesity-induced glucose intolerance. The regulatory importance of it may be related to the stimulation of PPARγ (peroxisome proliferator-activated receptor gamma), which mediates liver adipogenesis and lipogenesis [90,210]. Table 1 shows some osteokines and their effects. Figure 4 summarizes the relationship between the liver and bones.

In conclusion, although traditionally associated with bone metabolism, osteokines such as osteocalcin and osteopontin have emerged as influential regulators of energy metabolism and liver health. Their involvement in insulin sensitivity, inflammation, and lipid handling reveals a novel endocrine axis between bone and liver. Future studies are essential to clarify their exact role and therapeutic potential in MASLD, especially considering the frequent overlap between hepatic and skeletal complications in metabolic disorders.

## 7. Conclusions

Tissue-specific organokines play crucial roles in modulating MASLD, acting on interorgan axes (muscle-liver, bone-liver, kidney-liver, heart-liver, and adipose tissue-liver). While some (such as FGF-21, irisin, adiponectin, klotho, and osteocalcin) have protective effects, others (such as myostatin, fetuin-A, visfatin, and ANGPTL3) exacerbate insulin resistance, inflammation, and fibrosis. Dysfunction in these pathways reflects the systemic nature of MASLD, highlighting potential therapeutic targets for this condition. However, some cytokines, such as FGF-21 and myostatin, are produced in multiple tissues with distinct contextual effects, and there are context-dependent molecules, including IL-6 and IL-15. Dysregulation of these pathways reinforces the systemic nature of MASLD, highlighting the need for multiorgan therapeutic targets.

## 8. Future Perspectives

Future studies should aim to elucidate further the molecular pathways by which organokines, including hepatokines, adipokines, myokines, cardiokines, and osteokines, modulate the development and progression of MASLD. A detailed understanding of the intracellular signaling cascades involved in inflammation, fibrosis, steatosis, and systemic metabolic dysfunction will enable the identification of precise molecular targets for therapeutic intervention. These organokines could work as noninvasive biomarkers for early diagnosis, disease stratification, and treatment monitoring in MASLD. In the therapeutic approach, selective inhibition of pro-inflammatory organokines could mitigate hepatic inflammation and fibrosis. However, clinical trials are necessary to evaluate the safety, efficacy, and long-term outcomes of such interventions.

Another critical area for further developing MASLD research is the interaction between the gut microbiota and the liver. The gut microbiota, composed of bacteria, fungi, viruses, and archaea, along with its metabolites, can profoundly impact liver health, leading to the amelioration or exacerbation of liver diseases. Unhealthy changes to the gut microbiota can lead to intestinal barrier dysfunction and the displacement of microbial components, further leading to the development of MASLD [229]. The gut microbiota is also related to hepatocyte lipid metabolism and immune regulation, and recent research has linked interventions with gut microbiota to the brain, kidney, heart, adipose tissue, and muscle health. In this sense, future research endeavors must elucidate how new therapeutic avenues, including microbial-targeted therapies, dietary interventions, and metabolic surgery, can be used to treat and intervene against MASLD effectively [230,231,232].

Finally, given the multisystemic nature of MASLD, future therapeutic approaches should adopt a comprehensive and personalized strategy that includes modulation of organokine signaling, pharmacological interventions, lifestyle modification, and management of associated comorbidities. This integrative vision will be crucial in preventing disease progression and reducing long-term complications in individuals affected by the disease. However, clinical trials need to be performed to open the way for using these molecules as therapies for MASLD and other metabolic conditions.

## Figures and Tables

**Figure 1 ijms-26-11547-f001:**
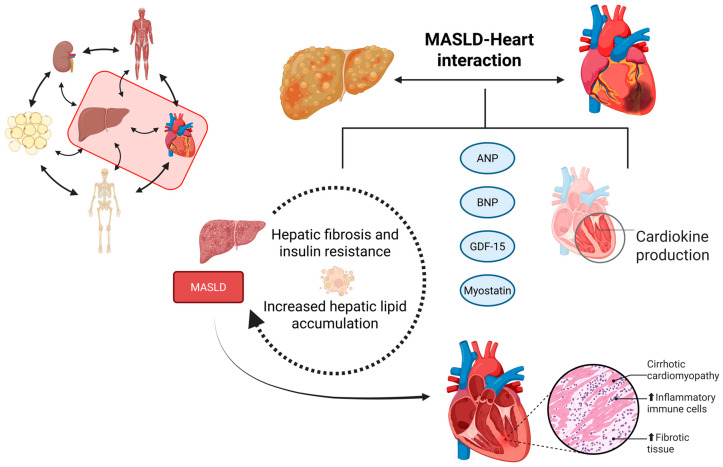
The relationship between the liver and the heart (Created in https://BioRender.com, last accessed on 20 November 2025). The heart interacts with the liver during MASLD progression through various pathways, most of which are associated with cardiokine production. Cardiokines can be harmful for MASLD progression, leading to increased liver fibrosis associated with insulin resistance, hepatic inflammation, and hepatic lipid accumulation. On the other hand, MASLD exacerbates cardiac fibrosis, leading to the infiltration of inflammatory cells, which in turn is associated with increased oxidative stress and the development of cirrhotic cardiomyopathy.

**Figure 2 ijms-26-11547-f002:**
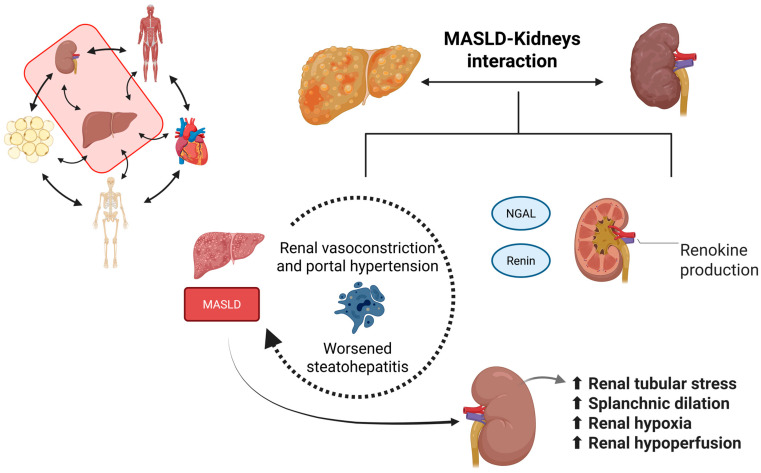
The relationship between the liver and the kidneys (Created in https://BioRender.com, last accessed on 20 November 2025). The liver and the kidneys interact with each other during health and disease. During MASLD, the kidneys’ health is worsened due to elevated renal tubular stress and an increase in the splanchnic vessels’ dilation, causing renal hypoxia and hypoperfusion. On the other hand, during MASLD, the kidneys produce renokines, such as NGAL and renin, which are associated with increased renal vasoconstriction and portal hypertension, as well as worsened steatohepatitis due to increased liver inflammation, oxidative stress, and lipid accumulation.

**Figure 3 ijms-26-11547-f003:**
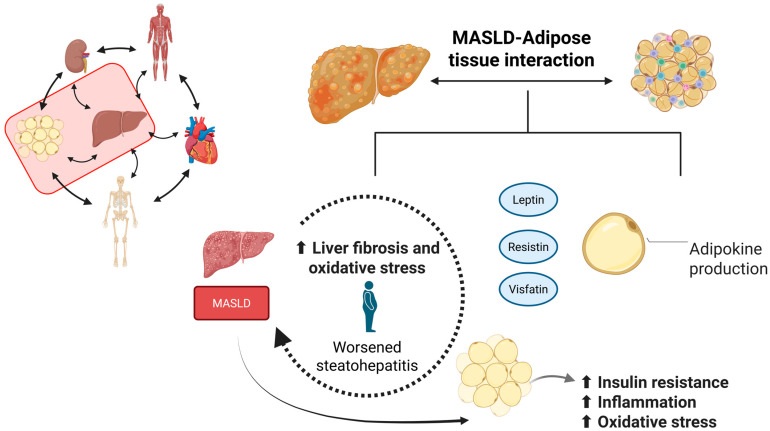
The relationship between the liver and the adipose tissue (Created in https://BioRender.com, last accessed on 20 November 2025). The adipose tissue plays a crucial role in maintaining the body’s homeostasis. When the liver is diseased, e.g., with MASLD, the adipose tissue primarily interacts with the organ by producing leptin, resistin, and visfatin, adipokines that worsen the prognosis of the liver disease. Leptin, resistin, and visfatin induce liver fibrosis and oxidative stress, accompanied by worsened steatohepatitis. On the other hand, MASLD worsens adipose tissue-related insulin resistance, inflammation, and oxidative stress.

**Figure 4 ijms-26-11547-f004:**
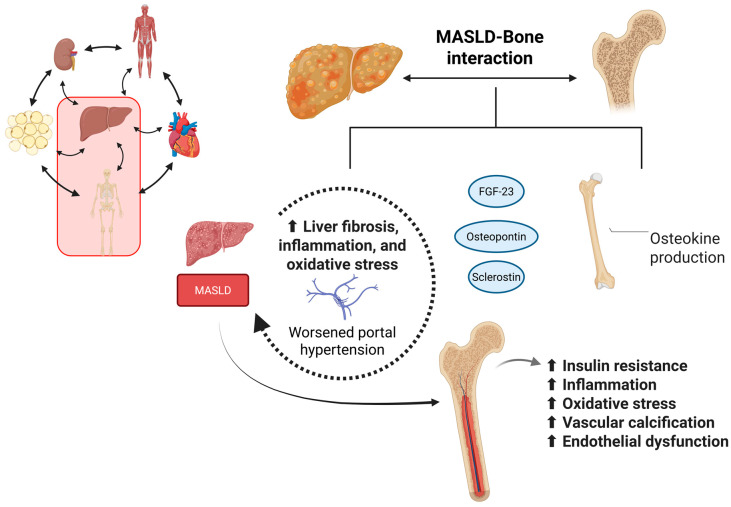
The relationship between the liver and the bones (Created in https://BioRender.com, last accessed on 20 November 2025). Bones also play a critical role in maintaining the body’s homeostasis. They are related to the production of several molecules, called osteokines, which are involved in various actions during the occurrence and progression of MASLD. Some aggressive osteokines, including FGF-23, osteopontin, and sclerostin, play a critical role in modulating liver fibrosis, inflammation, and oxidative stress during MASLD, thereby exacerbating portal hypertension as an outcome. On the other hand, MASLD induces insulin resistance, inflammation, oxidative stress, vascular calcification, and endothelial dysfunction through its interaction with osteokine production.

**Table 1 ijms-26-11547-t001:** Tissue-Derived Organokines Role in Metabolic Conditions Related to MASLD.

Tissue-Derived Cytokine	Condition/Stimulus for Release	Relations with MASLD	References
**Hepatokines**			
ANGPTL3	Liver lipid overload	Impairs lipid clearance; associated with dyslipidemia and liver fat	[48]
Activin E	↑ in obesity and MASLD	Regulatory molecule that prevents fatty acid influx into the liver	[49]
Fetuin-A	Liver stress and insulin resistance	Promotes insulin resistance and hepatic lipid accumulation	[50]
FGF-21	Fasting and metabolic stress	Enhances fatty acid oxidation; protective role in NAFLD	[51]
Selenoprotein P	Oxidative stress and liver dysfunction	Induces insulin resistance and hepatic inflammation	[52]
**Myokines**			
BDNF	↑ hepatic lipid oxidation	Regulates satiety; deficiency is associated with hyperphagia and ↑ hepatic fat deposition	[53,54,55]
FGF-21 (muscle-derived)	Metabolic stress and fasting	Improves lipid metabolism; protective in MASLD	[56]
Irisin	Physical exercise	Improves insulin sensitivity; reduces hepatic steatosis and inflammation	[57,58]
IL-6 (from muscle)	Acute exercise and chronic inflammation	Dual role: acute exercise-induced IL-6 is protective; chronically elevated levels may worsen hepatic inflammation	[59,60]
IL-15	Elevated serum levels in obesity; ↑ in NASH, it recruits NK lymphocytes to the liver (excess aggravates inflammation)	This process increases hepatic lipid accumulation and modulates macrophage infiltration in the liver	[61,62]
Myostatin	Muscle inactivity; metabolic disorders	This condition hinders muscle growth and is linked to insulin resistance and the accumulation of liver fat	[63,64]
**Cardiokines**			
ANP	Effective hypervolemia and atrial distension	Regulates liver glycogen metabolism and glucose homeostasis	[65]
BNP	Cardiac damage, acute myocardial infarction, and subclinical cardiac dysfunction in cirrhosis	Biomarker of volume overload and ↑ in cirrhotic cardiomyopathy	[66,67]
GDF-15	Tissue injury and oxidative stress	Linked to metabolic regulation and hepatic stress adaptation	[68,69]
IL-33	Cardiac stress and inflammation	May reduce liver inflammation and fibrosis in the early stages	[70]
Myostatin	Adipose tissue and muscle produce myostatin and worsen peripheral resistance; accumulation of uremic toxins	↑ Muscle atrophy; stimulates hepatic stellate cells (fibrosis)	[54,71]
Natriuretic peptides	Cardiac stretch and heart failure	Improve lipid metabolism; may protect against NAFLD progression	[72,73]
**Renokines**			
Erythropoietin	Hypoxia and anemia	Modulates insulin sensitivity; may reduce liver steatosis	[74,75]
Klotho	Kidney function regulation	Anti-inflammatory and antioxidant; protective effect in MAFLD	[76,77]
NGAL	Renal tubular stress	A marker of kidney stress, correlated with the severity of liver injury	[78,79]
Renin	↓ Renal perfusion; ↓ [Na^+^] in the distal tubule	Activation of the RAAS; worsening steatohepatitis	[73]
**Adipokines**			
Adiponectin	Caloric restriction and healthy adipose tissue	Anti-inflammatory and insulin-sensitizing; protective against liver steatosis and fibrosis	[80,81]
Leptin	Increased fat mass	Hyperleptinemia promotes hepatic inflammation and fibrosis; it is elevated in MAFLD	[82]
Resistin	Inflammation and obesity	↑ Insulin resistance is associated with hepatic lipid accumulation	[83,84]
Visfatin	Visceral obesity and inflammation	Activates pro-inflammatory pathways; ↑ production of TNF-α, IL-6, IL-1β; worsens steatohepatitis (NASH); ↑ liver fibrosis; ↑ oxidative stress	[85,86]
**Osteokines**			
FGF-23	Elevated levels in NAFLD	↑ Insulin resistance	[87,88,89]
NGAL	Known as a biomarker for acute kidney injury	Organogenesis and modulation of inflammation; elevated in metabolic diseases, including liver diseases	[90]
Osteocalcin	Bone remodeling, mechanical loading	↓ accumulation of lipids in the liver	[43,91]
Osteopontin	Inflammation and tissue injury	Promotes hepatic inflammation and fibrosis; elevated in MASLD and NASH	[92]
Sclerostin	Mechanical unloading of bone	Impairs insulin sensitivity; may contribute to metabolic dysfunction in liver disease	[93,94]

↓: decrease; ↑: increase; ANGPTL3: Angiopoietin-Like 3; ANP: Atrial Natriuretic Peptide; BNP: B-Type Natriuretic Peptide; BDNF: Brain-Derived Neurotrophic Factor; FGF-21: Fibroblast Growth Factor 21; FGF-23: Fibroblast Growth Factor 23; GDF-15: Growth Differentiation Factor 15; IL: Interleukin; MAFLD: Metabolic-associated fatty liver disease; MASLD: Metabolic dysfunction-associated steatotic liver disease; NAFLD: Non-Alcoholic Fatty Liver Disease; NGAL: Neutrophil gelatinase-associated lipocalin; NASH: Nonalcoholic steatohepatitis; RAAS: Renin-Angiotensin-Aldosterone System; TNF-α: Tumor Necrosis Factor-α.

## Data Availability

No new data were created or analyzed in this study. Data sharing is not applicable to this article.

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
