# Peer review of "Organokine-Mediated Crosstalk: A Systems Biology Perspective on the Pathogenesis of MASLD—A Narrative Review"

_ijms, 2025, doi:10.3390/ijms262311547_

Round 1

Reviewer 1 Report

Comments and Suggestions for Authors

The manuscript is a review focused on the presentation of the complex molecular interactions between the liver and other key organs in the pathophysiology of MASLD. The study is very interesting,  showing possible implications for diagnostic and treatment of the molecular events taking place in MASLD. Only a few minor issues need to be corrected ahead of publishing:

  1. I think it would be useful to present in a single phrase in the Introduction the former name of MASLD which was NAFLD. This article could be read by a wide audience, and not all the readers would know the recent (2023) change in the terminology of the disease.
  2. The authors mentioned in the manuscript in Line 183 and Figure 1 the term "selenoprotein" but they should be more precise and change with "selenoprotein P" because there are many selenoproteins not linked with MASLD.
  3. In the caption of Figure 3 it is mentioned "rinokines" please replace with "renokines"
  4. In Line 519 the authors have mentioned "holistic diagnostic", I would avoid the term "holistic" which has connotations linked to a less serious medicine.

Author Response

RESPONSE TO REVIEWERS' COMMENTS

Manuscript number: ijms-3908963 ― International Journal of Molecular Sciences

"Organokine-Mediated Crosstalk: A Systems Biology Perspective on the Pathogenesis of MASLD – A Narrative Review"

The authors of this document wish to express their deepest gratitude to the Editor-in-Chief and the Reviewer for their thorough and insightful evaluation of our manuscript. Their expert feedback has been invaluable in enhancing the quality of our work. We have carefully considered and diligently implemented each suggestion, which has significantly improved the manuscript. We have made substantial revisions to address the points raised. These noteworthy changes are marked mainly with YELLOW-highlighted text throughout the document for ease of reference. A note will be provided for the referee's attention, highlighting corrections in a different color. Additionally, we have prepared a detailed and comprehensive response to each comment and suggestion. This response is organized in a "point-by-point" format below, ensuring that every concern has been thoroughly addressed and explained. We sincerely appreciate the time and effort invested by the Editor-in-Chief and the Reviewer, and we believe their contributions have significantly strengthened the final version of our manuscript.

REVIEWER #1

General comment

The manuscript is a review focused on the presentation of the complex molecular interactions between the liver and other key organs in the pathophysiology of MASLD. The study is very interesting,  showing possible implications for diagnostic and treatment of the molecular events taking place in MASLD. Only a few minor issues need to be corrected ahead of publishing.

General response

Dear Erudite Reviewer, thank you for taking the time to revise our manuscript and allowing us to improve based on your valuable comments and suggestions. After addressing all your comments and suggestions regarding our manuscript text, we are confident that a significantly enhanced manuscript version has emerged. We are excited to resubmit the modified version for your perusal and reevaluation. Thank you for your brilliant insights, essential contributions, and feedback. You do have an eye for improvement. As a gesture of our utmost respect for you, we would like to provide you with a detailed and comprehensive point-by-point response to your comments below. Thank you once again for your time and patience in revising our article.

Comment #1

I think it would be useful to present in a single phrase in the Introduction the former name of MASLD which was NAFLD. This article could be read by a wide audience, and not all the readers would know the recent (2023) change in the terminology of the disease.

Response

Dear Erudite Reviewer, thank you for this precious comment and suggestion. You are correct that adding a sentence that delves into the first terminology of liver disease in our introduction would undoubtedly enhance its readability and content. Therefore, we included Lines 73-78 on Page 2 to reflect this correction and successfully enhance our manuscript’s quality based on your recommendations. In 1986, NAFLD was coined for a condition that would become the most prevalent liver disorder worldwide. However, over the last 3-4 years, specialists from around the world have come together to remove NAFLD and MASLD, given the global rise in the development of cardiometabolic risk factors. In this sense, the following reference has been added.

Portincasa, P.; Baffy, G. Metabolic dysfunction-associated steatotic liver disease: Evolution of the final terminology. Eur J Intern Med 2024, 124, 35–39, doi:10.1016/j.ejim.2024.04.013.

            Once again, thank you for your patience and keen eye for improvement. We greatly appreciate communicating with such an Esteemed Reviewer.

Comment #2

The authors mentioned in the manuscript in Line 183 and Figure 1 the term "selenoprotein" but they should be more precise and change with "selenoprotein P" because there are many selenoproteins not linked with MASLD.

Response

Dear Erudite Reviewer, thank you for this comment. You are entirely correct, and e agree with you that this part of the text should be more explicitly correlated with selenoprotein P. Therefore, we corrected the sentence in Lines 204-205 on Page 6 of the revised manuscript document to streamline the manuscript’s quality and readability based on your recommendations. Please note that, after the modifications suggested by Reviewer #2, the current Figure 1 no longer mentions selenoprotein P. We rebuilt all our figures with BioRender, and, especially, made the current Figure 1 a more nuanced explanation of the risk factors that drive MASLD occurrence. I hope you can understand that. Please encounter the new and revised Figure 1 on Page 7 of the revised manuscript document.

            Again, thank you for your attention to detail. It is a true honor for us to communicate with you.

Comment #3

In the caption of Figure 3 it is mentioned "rinokines" please replace with "renokines".

Response

Dear Erudite Reviewer, thank you for this comment and brilliant suggestion. We appreciate your attention to detail and eye for improvement. Following your suggestion, the caption of Figure 3 has been revised, and the correct terminology has been utilized throughout the entire sentence. Please find the revised caption in Lines 395-401 on Page 12 of the revised manuscript document. We hope that this modification will align better with your expectations for the publication of our manuscript. We look forward to hearing about your approval for its publication soon.

            Again, thank you for everything!

Comment #4

In Line 519 the authors have mentioned "holistic diagnostic", I would avoid the term "holistic" which has connotations linked to a less serious medicine.

Response

Dear Erudite Reviewer, you are entirely correct. Thank you for this comment. To improve our manuscript based on your findings, we have corrected the manuscript and removed the word “holistic” from its text. The phrase now reads “It underscores the importance of developing more effective diagnostic and therapeutic strategies that extend beyond the liver, targeting multiple axes of dysfunction simultaneously…” in the revised manuscript document in Lines 547-549 on Page 18. We hope that this modification will meet the rigorous standards of our manuscript.

            Thank you for your attention to detail and eye for improvement. It has been an honor communicating with you.

 I, the corresponding author of the manuscript "Organokine-Mediated Crosstalk: A Systems Biology Perspective on the Pathogenesis of MASLD – A Narrative Review" under the assigned ID ijms-3908963, on behalf of my coauthors, once again extend my heartfelt gratitude to the knowledgeable Editor-in-Chief and reviewers for their time and expertise in revising our manuscript. After we addressed their constructive and refined feedback and suggestions, a significantly improved manuscript version emerged. Undoubtedly, their insightful suggestions and feedback have significantly enhanced the quality of our manuscript. We respectfully are at the disposal of the Editor-in-Chief and the Reviewer to address any additional suggestions regarding our publication. Suppose you are satisfied with our newly refined and significantly improved version. In that case, we look forward to the acceptance of our article for publication in the prestigious International Journal of Molecular Sciences. Thank you once again for your time and expertise.

Reviewer 2 Report

Comments and Suggestions for Authors

The submitted review article is focused on MASLD, formerly known as NAFLD and the role of organokine-mediated crosslink between the liver and peripheral organs, such as skeletal muscle, heart, kidneys, bone, and adipose tissue. Having in mind, that MASLD is a public health issue, being not only a major cause of liver-related morbidity and mortality, but also an independent risk factor for the development of noncommunicable diseases, the topic of this article is important, especially regarding the fact that nowadays, there is no definite treatment for MASLD. However, there are too many narrative articles already published related to the role of multi-organ interactions in MASLD. Based on the fact that this is just a narrative review paper, the novelty and significance of the presented findings could not be seen. In addition, the quality of the paper has to be significantly improved in order to attract the attention in this filed. Some other issues also need to be addressed.

-Title is overambitious and does not reflect the content of this review paper in the proper way.

- Abstract: Based on the journal requirements abstract has to be revised. In the current version abstract is too general. Please add more findings.

-There are too many keywords.

-Introduction: State of the art should be better organized with the aim of the review paper.

-The intestinal microbiota plays a crucial role in multi-organ interactions in MASLD and key mechanisms include the gut-liver axis. Authors omitted to point out that in this review paper. 

-In addition, figures created using https://www.freepik.com/. that are too general. Authors should try to better present schematically the complex mechanism of MASLD disease.

-I recommend including a methodology section detailing how the literature sources were selected.

-Section 7. Is too general and most of the things are already mentioned.

Author Response

RESPONSE TO REVIEWERS' COMMENTS

Manuscript number: ijms-3908963 ― International Journal of Molecular Sciences

"Organokine-Mediated Crosstalk: A Systems Biology Perspective on the Pathogenesis of MASLD – A Narrative Review"

The authors of this document wish to express their deepest gratitude to the Editor-in-Chief and the Reviewer for their thorough and insightful evaluation of our manuscript. Their expert feedback has been invaluable in enhancing the quality of our work. We have carefully considered and diligently implemented each suggestion, which has significantly improved the manuscript. We have made substantial revisions to address the points raised. These noteworthy changes are marked mainly with YELLOW-highlighted text throughout the document for ease of reference. A note will be provided for the referee's attention, highlighting corrections in a different color. Additionally, we have prepared a detailed and comprehensive response to each comment and suggestion. This response is organized in a "point-by-point" format below, ensuring that every concern has been thoroughly addressed and explained. We sincerely appreciate the time and effort invested by the Editor-in-Chief and the Reviewer, and we believe their contributions have significantly strengthened the final version of our manuscript.

REVIEWER #2

General comment

The submitted review article is focused on MASLD, formerly known as NAFLD and the role of organokine-mediated crosslink between the liver and peripheral organs, such as skeletal muscle, heart, kidneys, bone, and adipose tissue. Having in mind, that MASLD is a public health issue, being not only a major cause of liver-related morbidity and mortality, but also an independent risk factor for the development of noncommunicable diseases, the topic of this article is important, especially regarding the fact that nowadays, there is no definite treatment for MASLD. However, there are too many narrative articles already published related to the role of multi-organ interactions in MASLD. Based on the fact that this is just a narrative review paper, the novelty and significance of the presented findings could not be seen. In addition, the quality of the paper has to be significantly improved in order to attract the attention in this filed. Some other issues also need to be addressed.

General response

Dear Erudite Reviewer, thank you for taking the time to revise our manuscript and allowing us to improve based on your valuable comments and suggestions. After addressing all your comments and suggestions regarding our manuscript text, we are confident that a significantly enhanced manuscript version has emerged. We are excited to resubmit the modified version for your perusal and reevaluation. Thank you for your brilliant insights, essential contributions, and feedback. You do have an eye for improvement. As a gesture of our utmost respect for you, we would like to provide you with a detailed and comprehensive point-by-point response to your comments below. Thank you once again for your time and patience in revising our article.

Comment #1

Title is overambitious and does not reflect the content of this review paper in the proper way.

Response

Dear Erudite Reviewer, thank you for this comment and brilliant suggestion. We appreciate your commitment to ensuring that our manuscript is as accurate as possible. In light of this comment, we revised the manuscript’s title to “Organokine-Mediated Crosstalk: A Systems Biology Perspective on the Pathogenesis of MASLD – A Narrative Review,” which can be found in Lines 2-3 on Page 1 of the revised manuscript document. We hope that this new title aligns with your expectations.

            Again, thank you for your attention to detail and eye for improvement.

Comment #2

Abstract: Based on the journal requirements abstract has to be revised. In the current version abstract is too general. Please add more findings.

Response

Dear Erudite Reviewer, thank you for this comment and suggestion. You are entirely correct that our abstract would benefit from the addition of more findings about the intricate relationship between multi-organ interactions in MASLD. Therefore, we included Lines 51-62 on Page 2 of the revised manuscript, incorporating additional results and interpretations of our findings into the main text of the abstract. We appreciate your attention to detail and eye for improvement, and look forward to your positive response regarding the modifications we’ve made in our manuscript following your brilliant suggestions.

            Again, thank you for everything!

Comment #3

There are too many keywords.

Response

Dear Erudite Reviewer, thank you for bringing this to our attention. In response to your comment, we updated the keywords in Line 63 on Page 2 of the revised manuscript document. Thank you for your attention to detail and eye for improvement. Our manuscript has been significantly improved since we made the necessary corrections.

Comment #4

Introduction: State of the art should be better organized with the aim of the review paper.

Response

Dear Erudite Reviewer, thank you for this comment. We appreciate that you have brought this to our attention, and we corrected our manuscript accordingly in Lines 115-121 on Page 3 of the revised manuscript document. Our review critically synthesizes current evidence on organokine-mediated interorgan crosstalk in MASLD pathogenesis, focusing on molecular mechanisms, systemic metabolic effects, and potential diagnostic and therapeutic targets. It examines how various cytokines—hepatokines, myokines, adipokines, cardiokines, renokines, and osteokines—interact to influence insulin sensitivity, lipid metabolism, inflammation, and fibrosis, and discusses emerging biomarkers and intervention strategies. We hope this addition has significantly improved our manuscript.

            Again, thank you for your attention to detail and eye for improvement. It has been an honor to communicate with you.

Comment #5

The intestinal microbiota plays a crucial role in multi-organ interactions in MASLD and key mechanisms include the gut-liver axis. Authors omitted to point out that in this review paper.

Response

Dear Erudite Reviewer, thank you for this comment. We truly appreciate your commitment to ensuring that our manuscript has the most complete information possible. Therefore, we included Lines 580-590 on Page 19 to delve into the relationship between the gut microbiota and the liver, kidney, muscle, heart, and adipose tissue axes. We believe our manuscript has been updated from the resolution of this comment.

We appreciate your help!

Comment #6

In addition, figures created using https://www.freepik.com/. that are too general. Authors should try to better present schematically the complex mechanism of MASLD disease.

Response

We sincerely appreciate the reviewer’s insightful comment regarding the quality and specificity of the figures. In response, we have thoroughly redesigned all seven figures in the manuscript using BioRender (https://BioRender.com), a professional scientific illustration platform. This allowed us to create original, high-resolution, and conceptually refined schematic representations tailored to the focus and complexity of our review.

Figure 1 has been completely restructured to depict, in greater detail, the multifactorial pathogenesis of MASLD, including the interplay between hepatic lipid metabolism, insulin resistance, inflammation, and fibrosis. The new figure now incorporates key risk factors and their associated disturbances, in line with current literature and systems biology approaches.

Additionally, Figures 2–7 have also been updated to ensure visual coherence, accuracy, and improved scientific interpretability. Each figure now emphasizes specific pathways, mediators, and organokine interactions relevant to MASLD progression, aiming to facilitate reader understanding of these complex processes.

We believe that these comprehensive revisions substantially improve the visual and conceptual quality of the manuscript, aligning it with the reviewer’s recommendations and enhancing its value as a didactic and integrative resource for the field.

            Below, you will find the specific Page and Line numbers where the figures are located, along with their captions. Again, thank you for everything!

Figure 1 (Page 7); Caption in Lines 221-227 on Page 7.

Figure 2 (Page 9); Caption in Lines 278-284 on Page 9.

Figure 3 (Page 12); Caption in Lines 395-401 on Page 12.

Figure 4 (Page 14); Caption in Lines 467-473 on Page 14.

Figure 5 (Page 16); Caption in Lines 515-522 on Page 16.

Figure 6 (Page 17); Caption in Lines 539-546 on Page 17.

Figure 7 (Page 18); Caption in Lines 556-557 on Page 18.

Comment #7

I recommend including a methodology section detailing how the literature sources were selected.

Response

Dear Erudite Reviewer, thank you for this valuable comment and insightful suggestion. We appreciate your commitment to ensuring that our manuscript meets the highest criteria for quality and reproducibility. Therefore, we have included a methodology section as Appendix A in our revised manuscript document (Page 20). Since a methodology section is not mandatory for narrative reviews, and our manuscript has been structured without a methods section within its main text and sections, we opted to include the methods as an appendix to ensure the manuscript's optimal readability. Appendices are published in the main manuscript, and we are focused on ensuring that they are of the utmost importance for the quality of our manuscript.

            Again, thank you for your attention to detail and eye for improvement. Our manuscript has undergone significant improvements since we corrected it following your comments and suggestions. We are thankful for the opportunity to communicate with you.

Comment #8

Section 7. Is too general and most of the things are already mentioned.

Response

Dear Erudite Reviewer, thank you for this comment. We appreciate your suggestion to streamline the content of Section 7 of our revised manuscript. We did so, and the content is streamlined. To the best of our view, the section has been condensed after the removal of more than 45% of its overlapping content from the above sections. Truly, this section is necessary due to the content of Figures 6-7. However, the introduction for the figures’ content, which is the section title, has been streamlined. Please find the new Section 7 in Lines 530-557 on Pages 16-18 of the revised manuscript document.

            Again, thank you for your attention to detail and eye for improvement. It has been an honor to communicate with you during this critical peer review process.

I, the corresponding author of the manuscript "Organokine-Mediated Crosstalk: A Systems Biology Perspective on the Pathogenesis of MASLD – A Narrative Review" under the assigned ID ijms-3908963, on behalf of my coauthors, once again extend my heartfelt gratitude to the knowledgeable Editor-in-Chief and reviewers for their time and expertise in revising our manuscript. After we addressed their constructive and refined feedback and suggestions, a significantly improved manuscript version emerged. Undoubtedly, their insightful suggestions and feedback have significantly enhanced the quality of our manuscript. We respectfully are at the disposal of the Editor-in-Chief and the Reviewer to address any additional suggestions regarding our publication. Suppose you are satisfied with our newly refined and significantly improved version. In that case, we look forward to the acceptance of our article for publication in the prestigious International Journal of Molecular Sciences. Thank you once again for your time and expertise.

Round 2

Reviewer 2 Report

Comments and Suggestions for Authors

General comments

In compare to the previous version, authors have made some improvements. However, most of the issues are still present. Authors should again carefully revise manuscript in order to improve the overall quality of the manuscript. In the current version article does not meet the high requirements of this journal. Please see previous comments.

Special comments

-Based on the journal requirements abstract has to be revised. In the current version abstract is too long.

-Introduction: State of the art should be better organized

- Section 7. Should be improved or omitted. The text is still too general.

-The number of figures should be reduced.  

Author Response

RESPONSE TO REVIEWER COMMENTS

Manuscript number: ijms-3908963 ― International Journal of Molecular Sciences

"Organokine-Mediated Crosstalk: A Systems Biology Perspective on the Pathogenesis of MASLD – A Narrative Review"

We greatly appreciated the reviewer’s constructive feedback and recognized that, although several improvements had been made in the previous revision, there were still areas requiring further attention. We undertook a careful and thorough revision of the manuscript, addressing both the current and previous comments. As a result, we believe the overall quality, organization, and clarity of the manuscript have been significantly improved. We have ensured that the revised version better aligns with the journal’s standards. Below, we provide a detailed response to each comment, with references to where changes were made.

Comment #1

Based on the journal requirements abstract has to be revised. In the current version abstract is too long.

Response

We acknowledged the reviewer’s concern regarding the length of the abstract. To comply with the journal’s formatting requirements, we revised the abstract by condensing the content and focusing on the key elements: the aim of the study, the main findings, and the conclusions. The revised abstract now presents a concise and informative summary of the work while remaining within the required word limit. Specifically, the previous abstract spanned Lines 27-62 on Pages 1-2, encompassing 35 lines. In the revised manuscript version, the abstract encompasses only 13 lines.

Changes made on page 1, lines 2740.

Comment #2

Introduction: State of the art should be better organized

Response

We agreed with the reviewer that the "State of the Art" paragraph needed better organization. In the revised manuscript, we restructured this part of the introduction to improve clarity and logical flow. The updated version begins with a review of foundational literature, followed by recent advances in the field, and concludes with a clear statement of the gap our study addresses. This reorganization helps contextualize our contribution more clearly within the existing body of research.

Changes made on pages 2-3, lines 88102.

Comment #3

Section 7. Should be improved or omitted. The text is still too general.

Response

We appreciated the reviewer’s feedback regarding Section 7. Although this section had been revised previously, we agreed that it remained too general and did not contribute substantively to the manuscript. As a result, we decided to remove Section 7 entirely in order to improve the overall focus, cohesion, and quality of the paper. This omission helped streamline the manuscript and eliminated redundancy.

Section 7 previously appeared on pages 16-18, lines 530–557 and has been removed.

Comment #4

The number of figures should be reduced.

Response

We acknowledged the reviewer’s point about the manuscript containing too many figures. In response, we carefully reviewed all visual content and identified opportunities to consolidate information. We identified two figures that presented related data and removed both, which were redundant or not essential to the main argument. This reduced the total number of figures (from 7 to 5) and improved the visual clarity of the manuscript, while still supporting the key points discussed in the text.

Changes made on pages 17 and 18

Final Response

We are grateful for the reviewer’s thoughtful and detailed comments, which helped us to substantially improve the manuscript. We have made careful revisions in response to each concern, and we believe the updated version is now more concise, better structured, and more aligned with the journal’s expectations. We hope that the reviewer will find the revised manuscript suitable for publication.